# Differentiable Blocks World: Qualitative 3D Decomposition by Rendering Primitives

**Tom Monnier**[1]    **Jake Austin**[2]    **Angjoo Kanazawa**[2]    **Alexei A. Efros**[2]    **Mathieu Aubry**[1]

[1]LIGM, Ecole des Ponts, Univ Gustave Eiffel          [2]UC Berkeley

## Abstract

Given a set of calibrated images of a scene, we present an approach that produces a simple, compact, and actionable 3D world representation by means of 3D primitives. While many approaches focus on recovering high-fidelity 3D scenes, we focus on parsing a scene into *mid-level* 3D representations made of a small set of textured primitives. Such representations are interpretable, easy to manipulate and suited for physics-based simulations. Moreover, unlike existing primitive decomposition methods that rely on 3D input data, our approach operates directly on images through differentiable rendering. Specifically, we model primitives as textured superquadric meshes and optimize their parameters from scratch with an image rendering loss. We highlight the importance of modeling transparency for each primitive, which is critical for optimization and also enables handling varying numbers of primitives. We show that the resulting textured primitives faithfully reconstruct the input images and accurately model the visible 3D points, while providing amodal shape completions of unseen object regions. We compare our approach to the state of the art on diverse scenes from DTU, and demonstrate its robustness on real-life captures from BlendedMVS and Nerfstudio. We also showcase how our results can be used to effortlessly edit a scene or perform physical simulations. Code and video results are available at www.tmonnier.com/DBW.

## 1   Introduction

Recent multi-view modeling approaches, building on Neural Radiance Fields [45], capture scenes with astonishing accuracy by optimizing a dense occupancy and color model. However, they do not incorporate any notion of objects, they are not easily interpretable for a human user or a standard 3D modeling software, and they are not useful for physical understanding of the scene. In fact, even though these approaches can achieve a high-quality 3D reconstruction, the recovered content is nothing but a soup of colorful particles! In contrast, we propose an approach that recovers textured primitives, which are compact, actionable, and interpretable.

More concretely, our method takes as input a collection of calibrated images of a scene, and optimizes a set of primitive meshes parametrized by superquadrics [1] and their UV textures to minimize a rendering loss. The approach we present is robust enough to work directly from a random initialization. One of its key components is the optimization of a transparency parameter for each primitive, which helps in dealing with occlusions as well as handling varying number of primitives. This notably requires adapting standard differentiable renderers to deal with transparency. We also show the benefits of using a perceptual loss, a total variation regularization on the textures and a parsimony loss favoring the use of a minimal number of primitives.

Our scene representation harks back to the classical Blocks World ideas [59]. An important difference is that the Blocks World-inspired approaches are typically bottom-up, leveraging low-level image features, such as edges [59], super-pixels [21], or more recently learned features [72, 32], to infer

37th Conference on Neural Information Processing Systems (NeurIPS 2023).

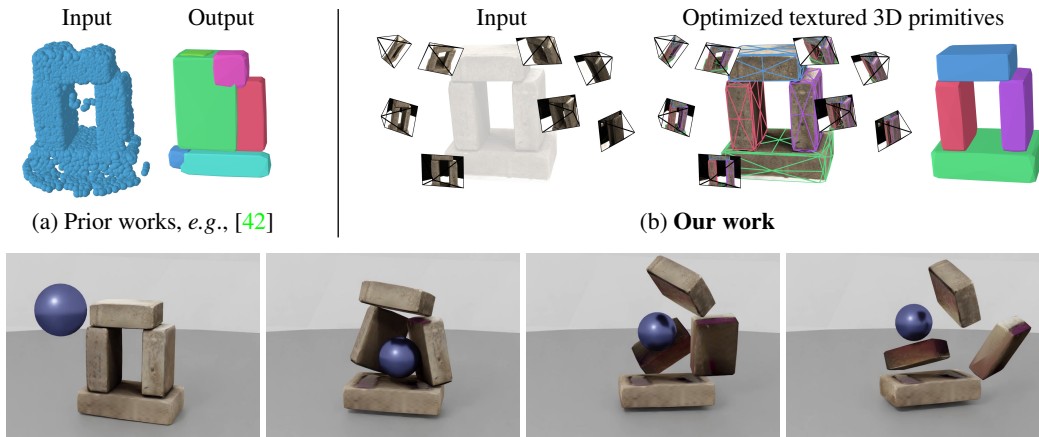

Figure 1: **Differentiable Blocks World. (a)** Prior works fit primitives to point clouds and typically fail for real data where ground-truth point clouds are extremely noisy and incomplete. **(b)** We propose using calibrated multi-view images instead and simultaneously tackle 3D decomposition and 3D reconstruction by rendering learnable textured primitives in a differentiable manner. **(c)** Such a textured decomposition is highly compact and user-friendly: it enables us to do physics-based simulations, *e.g.*, throwing a ball at the discovered primitives.

3D blocks. In contrast, we perform a direct top-down optimization of 3D primitives and texture using a rendering loss, starting from a random initialization in the spirit of analysis-by-synthesis. Unlike related works that fit primitives to 3D point clouds [3, 1, 68, 38, 71, 42, 43] (Figure 1a), our approach, dubbed *Differentiable Blocks World* (or DBW), does not require any 3D reconstruction *a priori* but instead operates directly on a set of calibrated input images, leveraging photometric consistency across different views (Figure 1b). This makes our approach more robust since methods based on 3D are very sensitive to noise in the reconstructions and have difficulties dealing with incomplete objects. Our setting is similar to existing NeRF-like approaches, but our model is able to recover a significantly more interpretable and parsimonious representation. In particular, such an interpretable decomposition allows us to easily play with the discovered scene, *e.g.*, by performing physics-based simulations (Figure 1c). Code and video results are available on our project webpage: www.tmonnier.com/DBW.

## 2   Related Work

**Scene decomposition into 3D primitives.**    The goal of understanding a scene by decomposing it into a set of geometric primitives can be traced back to the very fist computer vision thesis by Larry Roberts on *Blocks World* [59] in 1963. In it, Roberts shows a complete scene understanding system for a simple closed world of textureless polyhedral shapes by using a generic library of polyhedral block components. In the 1970s, Binford proposes the use of Generalized Cylinders as general primitives [3], later refined by Biederman into the recognition-by-components theory [2]. But applying these ideas to real-world image data has proved rather difficult.

A large family of methods does not consider images at all, instead focusing on finding primitives in 3D data. Building upon the classical idea of RANSAC [11], works like [4, 6, 62, 61, 39, 50, 57] accurately extract various primitive shapes (*e.g.*, planes, spheres and cylinders for [62, 61, 39]) from a point cloud. In particular, MonteBoxFinder [57] is a recent RANSAC-based system that robustly extracts cuboids from noisy point clouds by selecting the best proposals through Monte Carlo Tree Search. To avoid the need for RANSAC hyperparameter tuning while retaining robustness, Liu *et al*. [42] introduce a probabilistic framework dubbed EMS that recovers superquadrics [1]. Other methods instead leverage neural learning advances to robustly predict primitive decomposition from a collection of shapes (*e.g.*, ShapeNet [5]), in the form of cuboids [68], superquadrics [55, 53, 71], shapes from a small dictionary [38, 36] or learnable prototypical shapes [10, 54, 43]. However, they are typically limited to shapes of known categories and require perfect 3D data. More generally, the decomposition results of all 3D-based methods highly depend on the quality of the 3D input, which is always noisy and incomplete for real scenes. For a complete survey of 3D decomposition methods, we refer the reader to [28].

More recently, there has been a renewed effort to fit 3D primitives to various image representations, such as depth maps, segmentation predictions or low-level image features. Depth-based approaches [27, 12, 40, 18, 32] naturally associate a 3D point cloud to each image which is then used for primitive fitting. However, the resulting point cloud is highly incomplete, ambiguous and sometimes inaccurately predicted, thus limiting the decomposition quality. Building upon the single-image scene layout estimation [23, 24], works like [21, 37] compute cuboids that best match the predicted surface orientations. Finally, Façade [9], the classic image-based rendering work, leverages user annotations across multiple images with known camera viewpoints to render a scene with textured 3D primitives. In this work, we *do not* rely on 3D, depth, segmentation, low-level features, or user annotations to compute the 3D decomposition. Instead, taking inspiration from Façade [9] and recent multi-view modeling advances [69, 51, 45], our approach only requires calibrated views of the scene and directly optimizes textured primitives through photometric consistency in an end-to-end fashion. That is, we solve the 3D decomposition and multi-view stereo problems simultaneously.

**Multi-view stereo.** Our work can be seen as an end-to-end primitive-based approach to multi-view stereo (MVS), whose goal is to output a 3D reconstruction from multiple images taken from known camera viewpoints. We refer the reader to [22, 14] for an exhaustive review of classical methods. Recent MVS works can be broadly split into two groups.

Modular multi-step approaches typically rely on several processing steps to extract the final geometry from the images. Most methods [82, 16, 73, 74, 79, 20, 65], including the widely used COLMAP [63], first estimate depth maps for each image (through keypoint matching [63] or neural network predictions [73, 74, 79, 20, 65]), then apply a depth fusion step to generate a textured point cloud. Finally, a mesh can be obtained with a meshing algorithm [30, 34]. Other multi-step approaches directly rely on point clouds [15, 34] or voxel grids [64, 33, 26, 49]. Note that, although works like [26, 49] leverage end-to-end trainable networks to regress the geometry, we consider them as multi-step methods as they still rely on a training phase requiring 3D supervision before being applied to unknown sets of multi-view images. Extracting geometry through multiple steps involves careful tuning of each stage, thus increasing the pipeline complexity.

End-to-end approaches directly optimize a 3D scene representation using photometric consistency across different views along with other constraints in an optimization framework. Recent methods use neural networks to implicitly represent the 3D scene, in the form of occupancy fields [51], signed distance functions [77] or radiance fields, as introduced in NeRF [45]. Several works incorporate surface constraints in neural volumetric rendering to further improve the scene geometry [52, 76, 70, 8, 13], with a quality approaching that of traditional MVS methods. Other methods [17, 80, 19, 48] instead propose to leverage recent advances in mesh-based differentiable rendering [44, 29, 41, 7, 58, 35] to explicitly optimize textured meshes. Compared to implicit 3D representations, meshes are highly interpretable and are straightforward to use in computer graphic pipelines, thus enabling effortless scene editing and simulation [48]. However, all the above approaches represent the scene as a single mesh, making it ill-suited for manipulation and editing. We instead propose to discover the primitives that make up the scene, resulting in an interpretable and actionable representation. A concurrent work PartNeRF [67] introduces parts in NeRFs. However, only synthetic scenes with a single object are studied and the discovered parts mostly correspond to regions in the 3D space rather than interpretable geometric primitives.

## 3 Differentiable Blocks World

Given a set of $N$ views $\mathbf{I}_{1:N}$ of a scene associated with camera poses $\mathbf{c}_{1:N}$, our goal is to decompose the 3D scene into geometric primitives that best explain the images. We explicitly model the scene as a set of transparent superquadric meshes, whose parameters, texture and number are optimized to maximize photoconsistency through differentiable rendering. Note that compared to recent advances in neural volumetric representations [51, 45, 78], we *do not* use any neural network and directly optimize meshes, which are straightforward to use in computer graphic pipelines.

**Notations.** We use bold lowercase for vectors (*e.g.*, $\mathbf{a}$), bold uppercase for images (*e.g.*, $\mathbf{A}$), double-struck uppercase for meshes (*e.g.*, $\mathbb{A}$) and write $a_{1:N}$ the ordered set $\{a_1, \dots, a_n\}$.

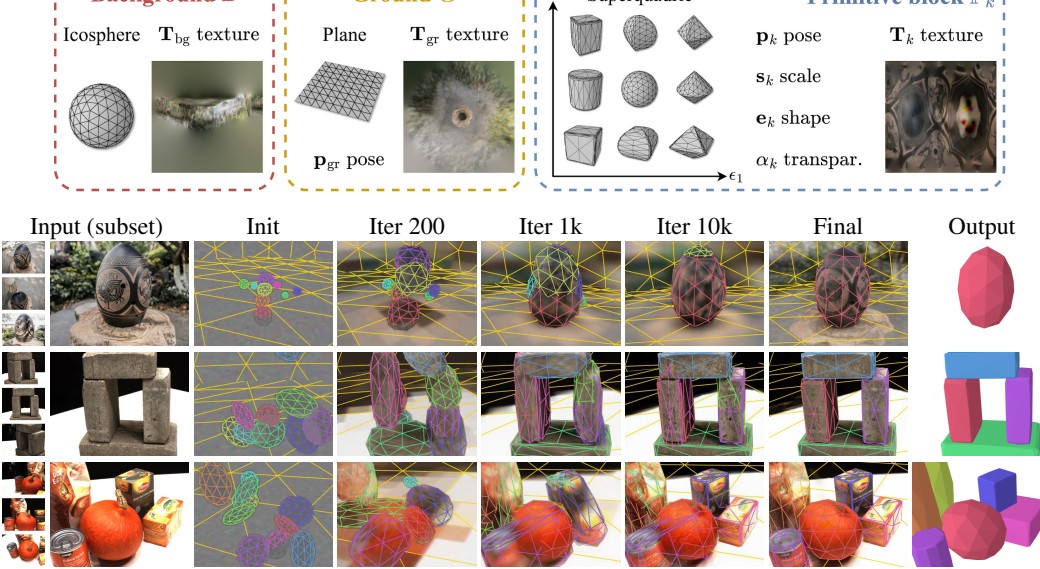

Figure 2: **Overview**. **(top)** We model the world as an explicit set of learnable textured meshes that are assembled together in the 3D space. **(bottom)** Starting from a random initialization, we optimize such a representation through differentiable rendering by photometric consistency across the different views.

## 3.1 Parametrizing a World of Blocks

We propose to represent the world scene as an explicit set of textured meshes positioned in the 3D space. Figure 2 summarizes our modeling and the parameters updated (top) during the optimization (bottom). Specifically, we model each scene as a union of primitive meshes: (i) an icosphere $\mathbb{B}$ modeling a background dome and centered on the scene, (ii) a plane $\mathbb{G}$ modeling the ground, and (iii) $K$ primitive blocks $\mathbb{P}_{1:K}$ in the form of superquadric meshes, where $K$ is fixed and refers to a maximum number of blocks. Unless mentioned otherwise, we arbitrarly use $K = 10$. We write the resulting scene mesh $\mathbb{B} \cup \mathbb{G} \cup \mathbb{P}_1 \cup \ldots \cup \mathbb{P}_K$.

The goal of the background dome is to model things far from the cameras that can be well approximated with a planar surface at infinity. In practice, we consider an icosphere with a fixed location and a fixed scale that is much greater than the scene scale. On the contrary, the goal of the planar ground and the blocks is to model the scene close to the cameras. We thus introduce rigid transformations modeling locations that will be updated during the optimization. Specifically, we use the 6D rotation parametrization of [83] and associate to each block $k$ a pose $\mathbf{p}_k = \{\mathbf{r}_k, \mathbf{t}_k\} \in \mathbb{R}^9$ such that every point of the block $\mathbf{x} \in \mathbb{R}^3$ is transformed into world space by $\mathbf{x}_{\text{world}} = \text{rot}(\mathbf{r}_k)\mathbf{x} + \mathbf{t}_k$, where $\mathbf{t}_k \in \mathbb{R}^3$, $\mathbf{r}_k \in \mathbb{R}^6$ and rot maps a 6D vector to a rotation matrix [83]. Similarly, we associate a rigid transformation $\mathbf{p}_{\text{gr}} = \{\mathbf{r}_{\text{gr}}, \mathbf{t}_{\text{gr}}\}$ to the ground plane. We next describe how we model variable number of blocks via transparency values and the parametrization of blocks' shape and texture.

**Block existence through transparency.** Modeling a variable number of primitives is a difficult task as it involves optimizing over a discrete random variable. Recent works tackle the problem using reinforcement learning [68], probabilistic approximations [55] or greedy algorithms [47], which often yield complex optimization strategies. In this work, we instead propose to handle variable number of primitive blocks by modeling meshes that are *transparent*. Specifically, we associate to each block $k$ a learnable transparency value $\alpha_k$, parametrized with a sigmoid, that can be pushed towards zero to change the effective number of blocks. Such transparencies are not only used in our rendering process to softly model the blocks existence and occlusions (Section 3.2), but also in regularization terms during our optimization, *e.g.*, to encourage parsimony in the number of blocks used (Section 3.3).

**Superquadric block shape.** We model blocks with superquadric meshes. Introduced by Barr in 1981 [1] and revived recently by [55], superquadrics define a family of parametric surfaces that exhibits a strong expressiveness with a small number of continuous parameters, thus making a good candidate for primitive fitting by gradient descent. More concretely, we derive a superquadric mesh

from a unit icosphere. For each vertex of the icosphere, its spherical coordinates $\eta \in [-\frac{\pi}{2}, \frac{\pi}{2}]$ and $\omega \in [-\pi, \pi]$ are mapped to the superquadric surface through the parametric equation [1]:

$$\Phi(\eta, \omega) = \begin{bmatrix} s_1 \cos^{\epsilon_1} \eta \cos^{\epsilon_2} \omega \\ s_2 \sin^{\epsilon_1} \eta \\ s_3 \cos^{\epsilon_1} \eta \sin^{\epsilon_2} \omega \end{bmatrix}, \tag{1}$$

where $\mathbf{s} = \{s_1, s_2, s_3\} \in \mathbb{R}^3$ represents an anisoptropic scaling and $\mathbf{e} = \{\epsilon_1, \epsilon_2\} \in \mathbb{R}^2$ defines the shape of the superquadric. Both $\mathbf{s}$ and $\mathbf{e}$ are updated during the optimization process. Note that by design, each vertex of the icosphere is mapped continuously to a vertex of the superquadric mesh, so the icosphere connectivity - and thus the icosphere faces - is transferred to the superquadric mesh.

**Texturing model.** We use texture mapping to model scene appearance. Concretely, we optimize $K + 2$ texture images $\{\mathbf{T}_{\text{bg}}, \mathbf{T}_{\text{gr}}, \mathbf{T}_{1:K}\}$ which are UV-mapped onto each mesh triangle using pre-defined UV mappings. Textures for the background and the ground are trivially obtained using respectively spherical coordinates of the icosphere and a simple plane projection. For a given block $k$, each vertex of the superquadric mesh is associated to a vertex of the icosphere. Therefore, we can map the texture image $\mathbf{T}_k$ onto the superquadric by first mapping it to the icosphere using a fixed UV map computed with spherical coordinates, then mapping the icosphere triangles to the superquadric ones (see supplementary material for details).

## 3.2 Differentiable Rendering

In order to optimize our scene parameters to best explain the images, we propose to leverage recent mesh-based differentiable renderers [41, 7, 58]. Similar to them, our differentiable rendering corresponds to the soft rasterization of the mesh faces followed by a blending function. In contrast to existing mesh-based differentiable renderers, we introduce the ability to account for transparency. Intuitively, our differentiable rendering can be interpreted as an alpha compositing of the transparent colored faces of the mesh. In the following, we write pixel-wise multiplication with $\odot$ and the division of image-sized tensors corresponds to pixel-wise division.

**Soft rasterization.** Given a 2D pixel location $\mathbf{u}$, we model the influence of the face $j$ projected onto the image plane with the 2D occupancy function of [7] that we modify to incorporate the transparency value $\alpha_{k_j}$ associated to this face. Specifically, we write the occupancy function as:

$$\mathcal{O}_j^{\text{2D}}(\mathbf{u}) = \alpha_{k_j} \exp\left(\min\left(\frac{\Delta_j(\mathbf{u})}{\sigma}, 0\right)\right), \tag{2}$$

where $\sigma$ is a scalar hyperparameter modeling the extent of the soft mask of the face and $\Delta_j(\mathbf{u})$ is the signed Euclidean distance between pixel $\mathbf{u}$ and projected face $j$, such that $\Delta_j(\mathbf{u}) < 0$ if pixel $\mathbf{u}$ is outside face $j$ and $\Delta_j(\mathbf{u}) \geq 0$ otherwise. We consider the faces belonging to the background and the ground to be opaque, *i.e.*, use a transparency of 1 for all their faces in the occupancy function.

**Blending through alpha compositing.** For each pixel, we find all projected faces with an occupancy greater than a small threshold at this pixel location, and sort them by increasing depth. Denoting by $L$ the maximum number of faces per pixel, we build image-sized tensors for occupancy $\mathbf{O}_\ell$ and color $\mathbf{C}_\ell$ by associating to each pixel the $\ell$-th intersecting face attributes. The color is obtained through barycentric coordinates, using clipped barycentric coordinates for locations outside the face. Different to most differentiable renderers and as advocated by [46], we directly interpret these tensors as an ordered set of RGBA image layers and blend them through traditional alpha compositing [56]:

$$\mathcal{C}(\mathbf{O}_{1:L}, \mathbf{C}_{1:L}) = \sum_{\ell=1}^{L} \left( \prod_{p<\ell} (1 - \mathbf{O}_p) \right) \odot \mathbf{O}_\ell \odot \mathbf{C}_\ell. \tag{3}$$

We found this simple alpha composition to behave better during optimization than the original blending function used in [41, 7, 58]. This is notably in line with recent advances in differentiable rendering like NeRF [45] which can be interpreted as alpha compositing points along the rays.

## 3.3 Optimizing a Differentiable Blocks World

We optimize our scene parameters by minimizing a rendering loss across batches of images using gradient descent. Specifically, for each image $\mathbf{I}$, we build the scene mesh as described in Section 3.1

and use the associated camera pose to render an image $\hat{\mathbf{I}}$ using the rendering process detailed in Section 3.2. We optimize an objective function defined as:

$$\mathcal{L} = \mathcal{L}_{\text{render}} + \lambda_{\text{parsi}}\mathcal{L}_{\text{parsi}} + \lambda_{\text{TV}}\mathcal{L}_{\text{TV}} + \lambda_{\text{over}}\mathcal{L}_{\text{over}} \,, \tag{4}$$

where $\mathcal{L}_{\text{render}}$ is a rendering loss between $\mathbf{I}$ and $\hat{\mathbf{I}}$, $\lambda_{\text{parsi}}, \lambda_{\text{TV}}, \lambda_{\text{over}}$ are scalar hyperparameters and $\mathcal{L}_{\text{parsi}}, \mathcal{L}_{\text{TV}}, \mathcal{L}_{\text{over}}$ are regularization terms respectively encouraging parsimony in the use of primitives, favoring smoothness in the texture maps and penalizing the overlap between primitives. Our rendering loss is composed of a pixel-wise MSE loss $\mathcal{L}_{\text{MSE}}$ and a perceptual LPIPS loss [81] $\mathcal{L}_{\text{perc}}$ such that $\mathcal{L}_{\text{render}} = \mathcal{L}_{\text{MSE}} + \lambda_{\text{perc}}\mathcal{L}_{\text{perc}}$. In all experiments, we use $\lambda_{\text{parsi}} = 0.01, \lambda_{\text{perc}} = \lambda_{\text{TV}} = 0.1$ and $\lambda_{\text{over}} = 1$. Figure 2 (bottom) shows the evolution of our renderings throughout the optimization.

**Encouraging parsimony and texture smoothness.** We found that regularization terms were critical to obtain meaningful results. In particular, the raw model typically uses the maximum number of blocks available to reconstruct the scene, thus over-decomposing the scene. To adapt the number of blocks per scene and encourage parsimony, we use the transparency values as a proxy for the number of blocks used and penalize the loss by $\mathcal{L}_{\text{parsi}} = \sum_k \sqrt{\alpha_k}/K$. We also use a total variation (TV) penalization [60] on the texture maps to encourage uniform textures. Given a texture map $\mathbf{T}$ of size $U \times V$ and denoting by $\mathbf{T}[u,v] \in \mathbb{R}^3$ the RGB values of the pixel at location $(u,v)$, we define:

$$\mathcal{L}_{\text{tv}}(\mathbf{T}) = \frac{1}{UV} \sum_{u,v} \left( \left\| \mathbf{T}[u+1,v] - \mathbf{T}[u,v] \right\|_2^2 + \left\| \mathbf{T}[u,v+1] - \mathbf{T}[u,v] \right\|_2^2 \right), \tag{5}$$

and write $\mathcal{L}_{\text{TV}} = \mathcal{L}_{\text{tv}}(\mathbf{T}_{\text{bg}}) + \mathcal{L}_{\text{tv}}(\mathbf{T}_{\text{gr}}) + \sum_k \mathcal{L}_{\text{tv}}(\mathbf{T}_k)$ the final penalization.

**Penalizing overlapping blocks.** We introduce a regularization term encouraging primitives to not overlap. Because penalizing volumetric intersections of superquadrics is difficult and computationally expensive, we instead propose to use a Monte Carlo alternative, by sampling 3D points in the scene and penalizing points belonging to more than $\lambda$ blocks, in a fashion similar to [54]. Following [54], $\lambda$ is set to 1.95 so that blocks could slightly overlap around their surface thus avoiding unrealistic floating blocks. More specifically, considering a block $k$ and a 3D point $\mathbf{x}$, we define a soft 3D occupancy function $\mathcal{O}_k^{\text{3D}}$ as:

$$\mathcal{O}_k^{\text{3D}}(\mathbf{x}) = \alpha_k \, \text{sigmoid} \left( \frac{1 - \Psi_k(\mathbf{x})}{\tau} \right), \tag{6}$$

where $\tau$ is a temperature hyperparameter and $\Psi_k$ is the superquadric inside-outside function [1] associated to the block $k$, such that $\Psi_k(\mathbf{x}) \leq 1$ if $\mathbf{x}$ lies inside the superquadric and $\Psi_k(\mathbf{x}) > 1$ otherwise. Given a set of $M$ 3D points $\Omega$, our final regularization term can be written as:

$$\mathcal{L}_{\text{over}} = \frac{1}{M} \sum_{\mathbf{x} \in \Omega} \max \left( \sum_{k=1}^K \mathcal{O}_k^{\text{3D}}(\mathbf{x}), \, \lambda \right). \tag{7}$$

Note that in practice, for better efficiency and accuracy, we only sample points in the region where blocks are located, which can be identified using the block poses $\mathbf{p}_{1:K}$.

**Optimization details.** We found that two elements were key to avoid bad local minima during optimization. First, while transparent meshes enable differentiability w.r.t. the number of primitives, we observed a failure mode where two semi opaque meshes model the same 3D region. To prevent this behavior, we propose to inject gaussian noise before the sigmoid in the transparencies $\alpha_{1:K}$ to create stochasticity when values are not close to the sigmoid saturation, and thus encourage values that are close binary. Second, another failure mode we observed is one where the planar ground is modeling the entire scene. We avoid this by leveraging a two-stage curriculum learning scheme, where texture maps are downscaled by 8 during the first stage. We empirically validate these two contributions in Section 4.3. We provide other implementation details in the supplementary material.

## 4 Experiments

### 4.1 DTU Benchmark

**Benchmark details.** DTU [25] is an MVS dataset containing 80 forward-facing scenes captured in a controlled indoor setting, where the 3D ground-truth points are obtained through a structured

Table 1: **Quantitative results on DTU [25].** We use the official DTU evaluation to report Chamfer Distance (CD) between 3D reconstruction and ground-truth, **best** results are highlighted. We also highlight the average number of primitives found (#P) in green (smaller than 10) or red (larger than 10). Our performances correspond to a single random run (random) and a run automatically selected among 5 runs using the minimal rendering loss (auto). We augment the best concurrent methods with a filtering step removing the ground from the 3D input.

| Method | Input | Chamfer Distance (CD) per scene | | | | | | | | | | Mean CD | Mean #P |
|---|---|---|---|---|---|---|---|---|---|---|---|---|---|
| | | S24 | S31 | S40 | S45 | S55 | S59 | S63 | S75 | S83 | S105 | | |
| EMS [42] | NeuS-mesh | 8.42 | 8.53 | 7.84 | 6.98 | 7.2 | 8.57 | 7.77 | 8.69 | 4.74 | 9.11 | 7.78 | 9.6 |
| EMS [42] | 3D GT | 6.77 | 5.93 | 3.36 | 6.91 | 6.52 | 3.50 | 4.72 | 7.08 | 7.25 | 6.10 | 5.82 | 7.4 |
| MBF [57] | NeuS-mesh | 3.97 | 4.28 | 3.56 | 4.76 | 3.33 | 3.92 | 3.63 | 5.58 | 5.3 | 6.07 | 4.44 | 53.5 |
| MBF [57] | 3D GT | 3.73 | 4.79 | 4.31 | 3.95 | 3.26 | 4.00 | 3.66 | 3.92 | 3.97 | **4.25** | 3.98 | 16.4 |
| **Ours (random)** | Image | 5.41 | **3.13** | 1.57 | 4.93 | 3.08 | 3.66 | **3.40** | 2.78 | 3.94 | 4.85 | 3.67 | **4.6** |
| **Ours (auto)** | Image | **3.25** | **3.13** | **1.16** | **3.02** | **2.98** | 2.32 | **3.40** | 2.78 | 3.43 | 5.21 | 3.07 | 5.0 |
| EMS [42] + filter | 3D GT | 6.32 | 4.11 | 2.98 | 4.94 | 4.26 | 3.03 | 3.60 | 5.44 | 3.24 | 4.43 | 4.23 | 8.3 |
| MBF [57] + filter | 3D GT | **3.35** | **2.95** | **2.61** | **2.19** | **2.53** | **2.47** | **1.97** | **2.60** | **2.60** | 3.27 | **2.65** | 29.9 |

light scanner. We evaluate on 10 scenes (S24, S31, S40, S45, S55, S59, S63, S75, S83, S105) that have different geometries and a 3D decomposition that is relatively intuitive. We use standard processing practices [77, 76, 8], resize the images to $400 \times 300$ and run our model with $K = 10$ on all available views for each scene (49 or 64 depending on the scenes). We use the official evaluation presented in [25], which computes the Chamfer distance between the ground-truth points and points sampled from the 3D reconstruction, filtered out if not in the neighborhood of the ground-truth points. We evaluate two state-of-the-art methods for 3D decomposition, EMS [42] and MonteboxFinder (MBF) [57], by applying them to the 3D ground-truth point clouds. We also evaluate them in a setup comparable to ours, where the state-of-the-art MVS method NeuS [70] is first applied to the multi-view images to extract a mesh, which is then used as input to the 3D decomposition methods. We refer to this input as "NeuS-mesh".

**Results.** We compare our Chamfer distance performances to these state-of-the-art 3D decomposition methods in Table 1. For each method, we report the input used and highlight the average number of discovered primitives #P in green (smaller than 10) or red (larger than 10). Intuitively, overly large numbers of primitives lead to less intuitive and manipulative scene representations. Our performances correspond to a single random run (random) and a run automatically selected among 5 runs using the minimal rendering loss (auto). We augment the best concurrent methods with a filtering step using RANSAC to remove the planar ground from the 3D input. Overall, we obtain results that are much more satisfactory than prior works. On the one hand, EMS outputs a reasonable number of primitives but has a high Chamfer distance reflecting bad 3D reconstructions. On the other hand, MBF yields a lower Chamfer distance (even better than ours with the filtering step) but it outputs a significantly higher number of primitives, thus reflecting over-decompositions.

Our approach is qualitatively compared in Figure 3 to the best EMS and MBF models, which correspond to the ones applied on the 3D ground truth and augmented with the filtering step. Because the point clouds are noisy and incomplete (see 360° renderings in our supplementary material), EMS and MBF struggle to find reasonable 3D decompositions: EMS misses some important parts, while MBF over-decomposes the 3D into piecewise planar surfaces. On the contrary, our model is able to output meaningful 3D decompositions with varying numbers of primitives and very different shapes. Besides, ours is the only approach that recovers the scene appearance (last column). Also note that it produces a complete 3D scene, despite being only optimized on forward-facing views.

### 4.2 Real-Life Data and Applications

We present qualitative results on real-life captures in Figure 4. The first row corresponds to the *Campanile* scene from Nerfstudio repository [66] and the last four rows correspond to BlendedMVS scenes [75] that were selected in [76]. We adapt their camera conventions to ours and resize the images to roughly $400 \times 300$. From left to right, we show a subset of the input views, a rendering overlaid with the primitive edges, the primitives, as well as two novel view synthesis results. For each scene, we run our model 5 times and automatically select the results with the minimal rendering loss. We set the maximum number of primitives to $K = 10$, except the last row where it is increased to

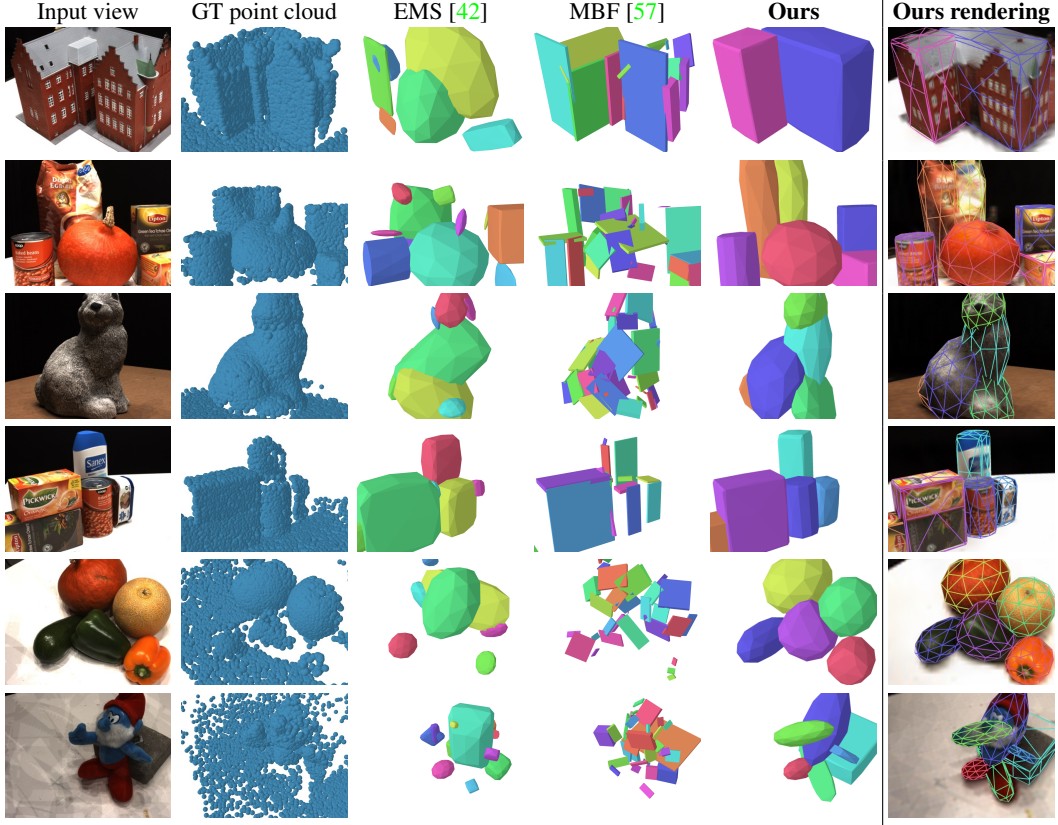

Figure 3: **Qualitative comparisons on DTU [25].** We compare our model to state-of-the-art methods (augmented with a preprocessing step to remove the 3D ground) which, unlike ours, find primitives in the ground-truth point cloud that is noisy and incomplete. Additionally, our approach is the only one able to capture the scene appearance (last column).

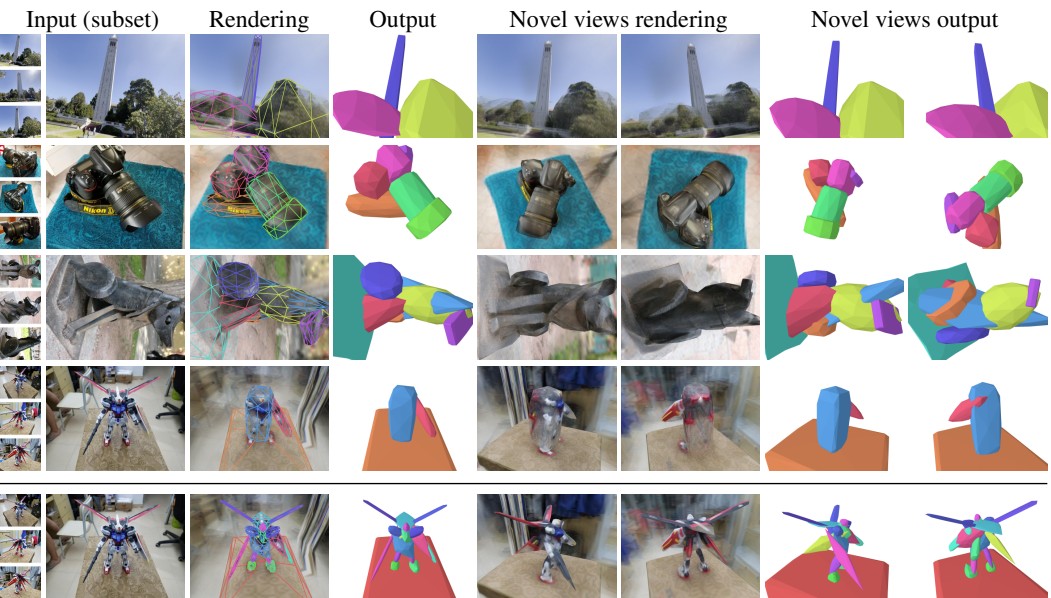

Figure 4: **Qualitative results on real-life data.** We run our default model ($K = 10$) on scenes from Nerfstudio [66] (first row) and BlendedMVS [75] (all other rows). The last row corresponds to results where the maximum number of primitives is increased to $K = 50$, yielding 17 effective primitives found.

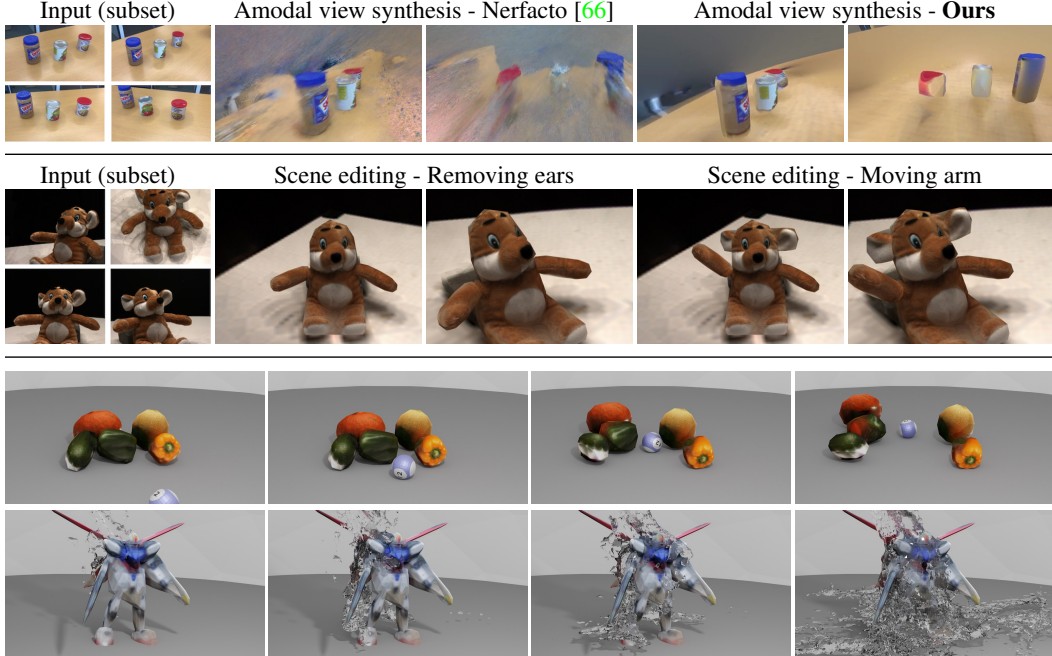

Figure 5: **Applications. (top)** Given a set of views constrained to limited viewpoint variations, we compare amodal view synthesis results using Nerfacto [66] and our approach. **(middle)** After optimization, we can easily modify the rendered scene by editing the different parts. **(bottom)** Our primitive-based representation enables straightforward physics-based simulations, such as throwing a ball at the objects or pouring water on the scene.

Table 2: **Ablation study on DTU [25].** We report metrics averaged over five runs: number of primitives (#P), Chamfer Distance (CD) and rendering metrics (PSNR in dB and SSIM, LPIPS in %). **Best** and second best are highlighted, #P variability is emphasized in green (smaller than 5) and red (larger than 5).

| Method | #P $\downarrow$ | CD $\downarrow$ | PSNR $\uparrow$ | SSIM $\uparrow$ | LPIPS $\downarrow$ |
|---|---|---|---|---|---|
| Complete model | $4.60 \pm 0.23$ | $\mathbf{3.63 \pm 0.23}$ | $\underline{20.5 \pm 0.2}$ | $\underline{73.5 \pm 0.6}$ | $\underline{23.9 \pm 0.5}$ |
| w/o $\mathcal{L}_{\text{parsi}}$ | $8.86 \pm 0.27$ | $\underline{3.65 \pm 0.22}$ | $\mathbf{20.6 \pm 0.1}$ | $\mathbf{73.7 \pm 0.4}$ | $\mathbf{23.2 \pm 0.4}$ |
| w/o $\mathcal{L}_{\text{over}}$ | $4.38 \pm 0.19$ | $3.80 \pm 0.30$ | $20.4 \pm 0.3$ | $73.2 \pm 0.7$ | $24.1 \pm 0.7$ |
| w/o curriculum | $4.66 \pm 0.30$ | $3.99 \pm 0.17$ | $20.4 \pm 0.2$ | $72.7 \pm 0.5$ | $24.5 \pm 0.4$ |
| w/o noise in $\alpha_{1:K}$ | $3.60 \pm 0.21$ | $4.13 \pm 0.28$ | $20.0 \pm 0.2$ | $72.0 \pm 0.6$ | $25.6 \pm 0.6$ |
| w/o $\mathcal{L}_{\text{TV}}$ | $4.04 \pm 0.18$ | $4.58 \pm 0.42$ | $19.7 \pm 0.3$ | $70.8 \pm 1.3$ | $26.5 \pm 1.2$ |
| w/o $\mathcal{L}_{\text{perc}}$ | $\mathbf{3.22 \pm 0.17}$ | $4.80 \pm 0.20$ | $19.7 \pm 0.1$ | $72.7 \pm 0.3$ | $40.0 \pm 0.4$ |

$K = 50$ due to the scene complexity. These results show that despite its simplicity, our approach is surprisingly robust. Our method is still able to compute 3D decompositions that capture both appearances and meaningful geometry on a variety of scene types. In addition, increasing the maximum number of primitives $K$ allows us to easily adapt the decomposition granularity (last row).

In Figure 5, we demonstrate other advantages of our approach. First, compared to NeRF-based approaches like Nerfacto [66] which only reconstruct visible regions, our method performs amodal scene completion (first row). Second, our textured primitive decomposition allows to easily edit the 3D scene (second row). Finally, our optimized primitive meshes can be directly imported into standard computer graphics softwares like Blender to perform physics-based simulations (bottom).

### 4.3 Analysis

**Ablation study.** In Table 2, we assess the key components of our model by removing one component at a time and computing the performance averaged over the 10 DTU scenes. We report the final number of primitives, Chamfer distance and rendering metrics. We highlight the varying number of primitives in green (smaller than 5) and red (larger than 5). Results are averaged over five runs,

Table 3: **Effect of hyperparameters on DTU [25].** We evaluate the influence of two key hyperparameters of our model: the maximum number of primitives $K$ (**left**) and the parsimony regularization $\lambda_{\text{parsi}}$ (**right**).

| Method | #P↓ | CD↓ | PSNR↑ | SSIM↑ | LPIPS↓ |
|---|---|---|---|---|---|
| $K = 10$ (default) | 4.60 | 3.63 | 20.5 | 73.5 | 23.9 |
| $K = 25$ | 7.00 | 3.58 | 21.0 | 74.6 | 22.5 |
| $K = 50$ | 9.26 | 3.52 | 20.9 | 74.7 | 22.8 |

| Method | #P↓ | CD↓ |
|---|---|---|
| $\lambda_{\text{parsi}} = 0.001$ | 7.44 | 3.61 |
| $\lambda_{\text{parsi}} = 0.01$ (default) | 4.60 | 3.63 |
| $\lambda_{\text{parsi}} = 0.1$ | 1.30 | 6.88 |

(a) Missing parts     (b) Unnatural decomposition     (c) Parsimony/fidelity trade-off

Figure 6: **Failure cases.** We show typical failure cases of our approach. All models are optimized with $K = 10$ except the rightmost model which is optimized with $K = 50$. See text for details.

we report the means and standard deviations. Overall, each component except $\mathcal{L}_{\text{parsi}}$ consistently improves the quality of the 3D reconstruction and the renderings. $\mathcal{L}_{\text{parsi}}$ successfully limits the number of primitives (and thus, primitive duplication and over-decomposition) at a very small quality cost.

**Influence of $K$ and $\lambda_{\text{parsi}}$.** In Table 3, we evaluate the impact of two key hyperparameters of our approach, namely the maximum number of primitives $K$ and the weight of the parsimony regularization $\lambda_{\text{parsi}}$. Results are averaged over the 10 DTU scenes for 5 random seeds. First, we can observe that increasing $K$ slightly improves the reconstruction and rendering performances at the cost of a higher effective number of primitives. Second, these results show that $\lambda_{\text{parsi}}$ directly influences the effective number of primitives found. When $\lambda_{\text{parsi}} = 0.1$, this strong regularization limits the reconstruction to roughly one primitive, which dramatically decreases the performances. When $\lambda_{\text{parsi}}$ is smaller, the effective number of primitives increases without significant improvements in the reconstruction quality.

**Limitations and failure cases.** In Figure 6, we show typical failure cases of our approach. First, for a random run, we may observe bad solutions where parts of the geometry are not reconstructed (Figure 6a). This is mainly caused by the absence of primitives in this region at initialization and our automatic selection among multiple runs alleviates the issue, yet this solution is computationally costly. Note that we also tried to apply a Gaussian kernel to blur the image and propagate gradients farther, but it had little effect. Second, our reconstructions can yield unnatural decompositions as illustrated in Figure 6b, where tea boxes are wrongly split or a single primitive is modeling the bear nose and the rock behind. Finally, in Figure 6c, we show that increasing $K$ from 10 (left) to 50 (right) allows us to trade-off parsimony for reconstruction fidelity. However, while this provides a form of control over the decomposition granularity, the ideal decomposition in this particular case does not seem to be found: the former seems to slightly under-decompose the scene while the latter seems to over-decompose it.

## 5   Conclusion

We present an end-to-end approach that successfully computes a primitive-based 3D reconstruction given a set of calibrated images. We show its applicability and robustness through various benchmarks, where our approach obtains better performances than methods leveraging 3D data. We believe our work could be an important step towards more interpretable multi-view modeling.

## Acknowledgments and Disclosure of Funding

We thank Cyrus Vachha for help on the physics-based simulations; Antoine Guédon, Romain Loiseau for visualization insights; François Darmon, Romain Loiseau, Elliot Vincent for manuscript feedback. This work was supported in part by ANR project EnHerit ANR-17-CE23-0008, gifts from Adobe and HPC resources from GENCI-IDRIS (2022-AD011011697R2, 2022-AD011013538). MA was supported by ERC project DISCOVER funded by the European Union's HorizonEurope Research and Innovation programme under grant agreement No. 101076028. Views and opinions expressed are however those of the authors only and do not necessarily reflect those of the European Union. Neither the European Union nor the granting authority can be held responsible for them.

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

# Supplementary Material for Differentiable Blocks World: Qualitative 3D Decomposition by Rendering Primitives

In this supplementary document, we provide additional results (Appendix A), details on the DTU benchmark (Appendix B) as well as implementation details (Appendix C), including design and optimization choices.

## A  Additional Results

**Videos for view synthesis, physical simulations and amodal completion.**  We present additional results in the form of videos at our project webpage: www.tmonnier.com/DBW. Videos are separated in different sections depending on the experiment type. First, we provide view synthesis videos (rendered using a circular camera path), further outlining the quality of both our renderings and our primitive-based 3D reconstruction. Second, we include videos for physics-based simulations. Such simulations were produced through Blender by simply uploading our output primitive meshes. Note that for modeling primitive-specific motions in Blender (*e.g.*, in our teaser figure), primitives should not overlap at all, thus requiring a small preprocessing step to slightly move the primitives for a clear separation. Because each primitive is its own mesh, this operation is easily performed within Blender. Finally, we provide video results where we perform scene editing and compare our amodal view synthesis results to the ones of Nerfacto introduced in Nerfstudio [66]. Models for amodal synthesis are optimized on a homemade indoor scene built from a forward-facing capture only. We use Nerfstudio for data processing and data convention.

**Rendering comparison with SOTA MVS.** For completeness, we provide the rendering performances of NeRF [45], a SOTA MVS method that does not predict multiple parts. In Table 4, we compare PSNR for our approach and NeRF using the results reported in [76] on the intersected set of 6 DTU scenes.

Table 4: **PSNR comparison on DTU.**

| Method | S24 | S40 | S55 | S63 | S83 | S105 | Mean |
|---|---|---|---|---|---|---|---|
| NeRF | 26.2 | 26.8 | 27.6 | 32.0 | 32.8 | 32.1 | 29.6 |
| Ours | 19.1 | 21.8 | 22.6 | 23.4 | 22.3 | 20.8 | 21.7 |

## B  DTU Benchmark

In Figure 7, we show for each scene a subset of the input images as well as 360° renderings of the GT point clouds obtained through a structured light scanner. To compute performances, we use a Python version of the official evaluation: https://github.com/jzhangbs/DTUeval-python.

## C  Implementation Details

**Icosphere and superquadric UV mapping.**  We use spherical coordinates that we correct to build our texture mapping for the unit icosphere. Figure 8 shows our process with an example. Specifically, we retrieve for each vertex its spherical coordinates $\eta \in [-\frac{\pi}{2}, \frac{\pi}{2}]$ and $\omega \in [-\pi, \pi]$ which are linearly mapped to the UV space $[0, 1]^2$. Because such parametrization presents discontinuities and strong triangle deformations at the poles, we perform two corrections. First, we fix discontinuities by copying the border pixels involved (using a circular padding on the texture image) and introducing new 2D vertices such that triangles do not overlap anymore. Second, we avoid distorted triangles at the poles by creating for each triangle, a new 2D vertex positioned in the middle of the other two vertices. As detailed in the main paper, we derive a superquadric mesh from a unit icosphere in such a way that each vertex of the icosphere is continuously mapped to the superquadric vertex. As a result, the texture mapping defined for the icosphere is directly transferred to our superquadric meshes without any modification.

**Design choices.**  Except constants related to the world scale, orientation and position in the 3D space w.r.t. to the known cameras, all our experiments share the same design choices. Specifically, all the following design choices are defined for a canonical 3D scene assumed to be centered and

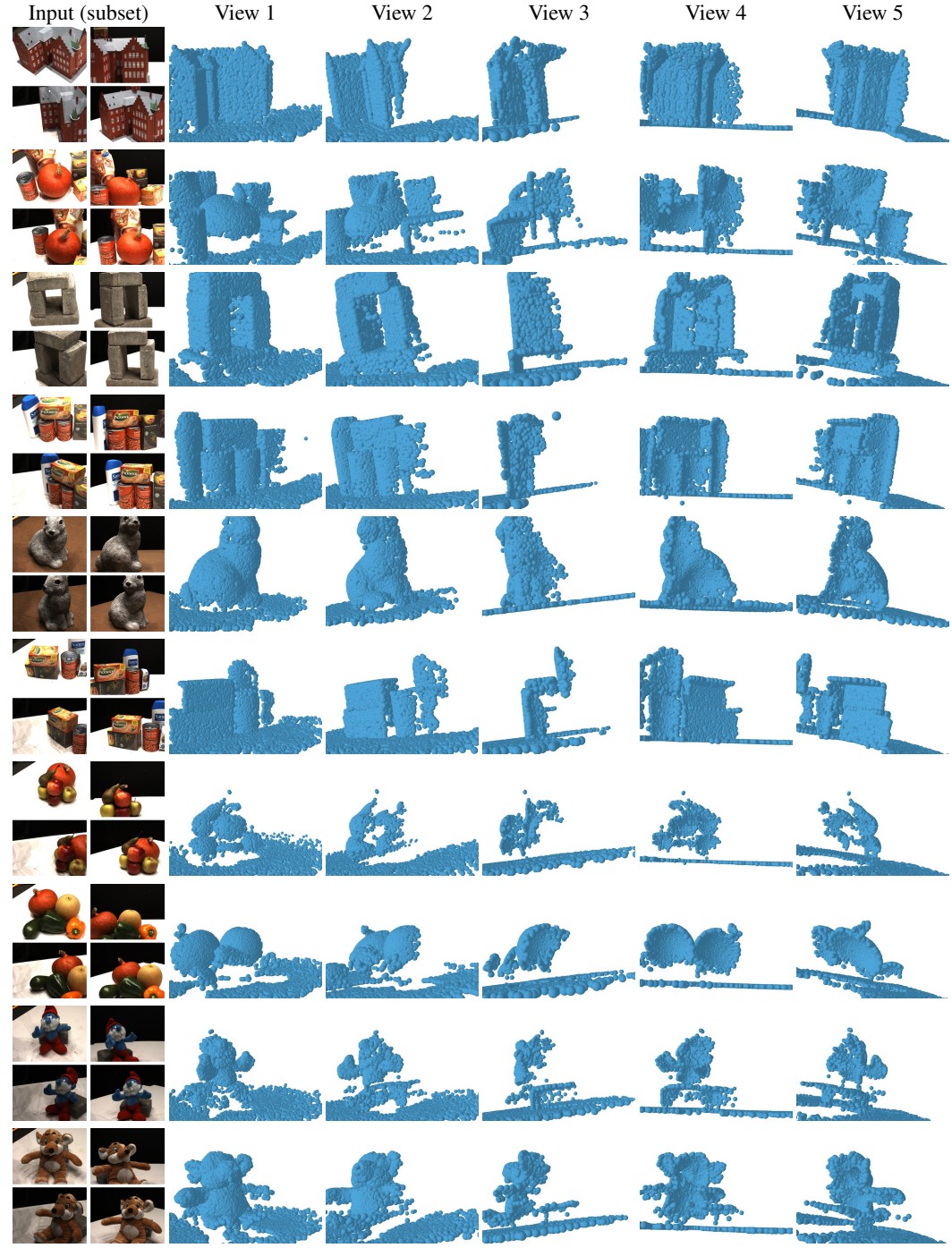

Figure 7: **DTU [25] scenes with ground-truth.** We show a subset of the input images as well as renderings of the GT point clouds. From top to bottom, scenes are: S24, S31, S40, S45, S55, S59, S63, S75, S83, S105.

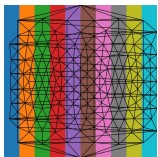 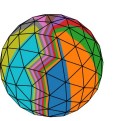  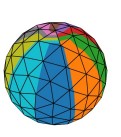 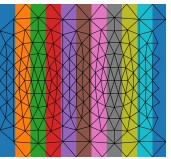 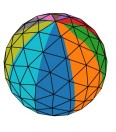



(a) Raw spherical coordinates      (b) Fixing border discontinuities      (c) Fixing distortions at the poles



Figure 8: **Our icosphere UV-mapping.** We illustrate different UV parametrizations using raw spherical coordinates **(a)** as well as our modified coordinates to fix discontinuities **(b)** and to prevent distortions at the poles **(c)**. For each parametrization, we show the texture image with the face edges representing the UV-mapping as well as a rendering example of the associated icosphere.

mostly contained in the unit cube, with a y-axis orthogonal to the ground and pointing towards the sky. We roughly estimate the scene-specific constants related to the world scale and pose (through coarse visual comparisons or using the camera locations), and apply them to our final scene model to account for the camera conventions.

The background corresponds to a level-2 icosphere (320 faces), the ground plane is subdivided into 128 uniform faces (for visual purposes) and superquadric meshes are derived from level-1 icospheres (80 faces). The scale for the background and the ground is set to 10. The ground is initialized perpendicular to the y-axis and positioned at $[0, -0.9, 0]$. The poses of our primitive blocks are initialized using a Gaussian distribution for the 3D translation and a random 6D vector for the rotation such that rotations are uniformly distributed on the unit sphere. We parametrize their scale with an exponential added to a minimum scale value of 0.2 to prevent primitives from becoming too small. These scales are initialized with a uniform distribution in $[0.5, 1.5]$ and multiplied by a constant block scale ratio of 0.25 to yield primitives smaller than the scene scale. The superquadric shape parameters are implemented with a sigmoid linearly mapped in $[0.1, 1.9]$ and are initialized at 1 (thus corresponding to a raw icosphere). Transparency values are parametrized with a sigmoid and initialized at 0.5. All texture images have a size of $256 \times 256$, are parametrized using a sigmoid and are initialized with small Gaussian noises added to gray images.

**Optimization details.** All our experiments share the same optimization details. We use Pytorch3D framework [58] to build our custom differentiable rendering process and use the default hyperparameter $\sigma = 10^{-4}$. Our model is optimized using Adam [31] with a batch size of 4 for roughly a total of 25k iterations. We use learning rates of 0.05 for the texture images and 0.005 for all other parameters, and divide them by 10 for the last 2k iterations. Following our curriculum learning process, we optimize the model for the first 10k iterations by downsampling all texture images by 8. Then, we optimize using the full texture resolution during the next 10k iterations. Finally, to further increase the rendering quality, we threshold the transparency values at 0.5 to make them binary, remove regularization terms related to transparencies (*i.e.*, $\mathcal{L}_{\text{parsi}}$ and $\mathcal{L}_{\text{over}}$), divide the weights for the other terms $\mathcal{L}_{\text{perc}}$ and $\mathcal{L}_{\text{TV}}$ by 10, decrease the smoothness rendering parameter $\sigma$ to $5 \times 10^{-6}$ and finetune our model for the final 5k iterations. In particular, this allows the model to output textures that are not darken by non-binary transparencies. During the optimization, we systematically kill blocks reaching a transparency lower than 0.01 and at inference, we only show blocks with a transparency greater than 0.5. Similar to [54], we use $\lambda = 1.95$ and $\tau = 0.005$ in our overlap penalization.

**Computational cost.** Optimizing our model on a scene roughly takes 4 hours on a single NVIDIA RTX 2080 Ti GPU. Since MBF [57] and EMS [42] directly operate on the 3D point cloud without computing textures, they are much faster and compute primitives in a couple of minutes. To get comparable timings however, we have to account for a method that computes 3D point clouds from the calibrated images, which is typically longer depending on the method. For example, we report MBF and EMS results using the mesh extracted from NeUS [70], which typically takes 14 hours to converge on a single DTU scene.

