# OpenReview forum: "Differentiable Blocks World: Qualitative 3D Decomposition by Rendering Primitives"
_NeurIPS.cc/2023/Conference — NeurIPS 2023 poster_

### Official Review · Reviewer_3Dgw · 2023-06-27

**Soundness:** 3 good
**Presentation:** 4 excellent
**Contribution:** 2 fair
**Rating:** 5
**Confidence:** 5

**Summary:**

The paper presents a method for producing simple "blocks world" 3D geometry for a scene from posed RGB images. Compared with multi-view 3D reconstruction approaches that produce point clouds or nerfs, the emphasis is on producing a simple, robust and (somewhat) object-aligned 3D representation. The representation used here is a collection of superquadric blobs. The main technical contribution is in the adaptation of differentiable rendering techniques to collections of superquadrics with support for transparency. Experiments show fitting various numbers of superquadrics to standard multi-view datasets such as DTU. The results show that the method can fit scenes with multiple sub-objects with reasonable quality, but is limited in the details it can reconstruct due to a limited number of superquadrics (usually ~10, one example shows 17).

**Strengths:**

The paper is very well written and the main concepts are easy to understand. Relevant related work is discussed in detail.

The approach is simple and appealing: given posed images, fit a set of textured primitives such that they reconstruct the images. It is a purely optimization-based approach without a learned prior, so in theory it should apply equally well across any image collection. The main technical novelty beyond the high-level concept is an adaptation of the soft rasterizer idea to handle transparent layers of superquadrics, which may be useful in other contexts.

The work could encourage others to investigate these sorts of "blocks world" representations.

**Weaknesses:**

My main concern with the paper is the results. The quality of the fits is pretty good for the number of superquadrics used, but still coarse, and the texture map can only do so much if the geometric complexity of the shape is higher than the superquadrics can handle (e.g., pitched roof of the building, top row fig. 3). The most impressive result is the gundam figure with 17 primitives (fig. 4), and the system apparently produces tight fits only for actual "blocks world" scenes (fig. 1).

I understand that the method is not targeted at photorealistic reconstruction. However, the usual criticism of "blocks world"-style approaches is that they are unrealistic, because the world isn't actually made out of blocks. The fidelity of the results of this method seem to confirm this criticism, which is a bit disappointing.

The limitations section suggests that the method is prone to bad local minima. This is a reasonable limitation but it is not surprising; it seems like an obvious limitation of this sort of optimization. Standard techniques such as coarse-to-fine optimization, or primitive splitting, were apparently not explored (unlike, for example, 3D Gaussian Splatting, Kerbl, et al., SIGGRAPH 2023). The paper states that K=10 was chosen for all experiments (l. 239) but the reader is left to wonder why this (small) number of primitives was chosen.

More minor: the adaptation of soft rasterizer to multiple layers seems like it could get slow for many layers, but this limitation is also not discussed.

**Questions:**

How was the number K=10 chosen? What happens if K=100, or K=1000? More generally, what is limiting the detail level of the fits?

Why does the model get caught in bad local minima (in more detail)? Would just increasing the number of restarts avoid the minima?

Are the parameters (sigma) of the soft rasterization adjusted on a schedule during optimization, similar to the original soft rasterizer paper? What is the effect, if any, of the soft rasterizer approximation on the optimization (i.e., what is the effect of varying sigma)?

How does the model behave for different values of the parsimony parameter? Is there an ablation showing the effect of that parameter?



**Limitations:**

As mentioned above, it would be nice to have a more detailed description of the limits of the optimization. Just saying it gets caught in bad local minima isn't all that helpful. What is the cause of these minima? Why doesn't the soft rasterization and the transparency handling smooth out the loss landscape sufficiently?

---

> ### Author Rebuttal · Authors · 2023-08-09
>
> We thank the reviewer for their valuable feedback. In the following, we address the main concerns raised in this review. We will incorporate relevant suggestions in the revised version.
>
> ***Weaknesses: "The system apparently produces tight fits only for actual "blocks world" scenes",  "The usual criticism of "blocks world"-style approaches is that they are unrealistic, because the world isn't actually made out of blocks. The fidelity of the results of this method seem to confirm this criticism, which is a bit disappointing."***\
> A: While we believe our results show that we can represent objects and scenes of significant complexity with a reasonable accuracy (using multiple superquadrics, e.g., Fig. 4), we agree that a "blocks world" approach such as ours will never be able to represent fine geometric details or extremely complex objects. However, while there are plenty of works that aim at photorealism (e.g., NeRF), they typically do not output an actionable representation, for which some sort of "mid-level representations" like blocks are necessary, and which is useful for some applications (e.g., we demonstrate physical simulation, scene manipulation and amodal view synthesis). We tackle this less explored problem of decomposing a scene into actionable primitives and we believe we are the first to do so from images alone.
>
> ***Weaknesses: "The method is prone to bad local minima. [...] Standard techniques such as coarse-to-fine optimization, or primitive splitting, were apparently not explored"***\
> A: Our curriculum learning can actually be seen as a coarse-to-fine optimization, where we gradually increase the resolution of the texture maps. We found this procedure to be particularly effective in solving the bad local minima where the entire scene is approximated with the textured ground plane. Refining the 3D reconstruction by splitting primitives or adding smaller primitives during the course of the optimization is an interesting path of research but is quite different from our main objective to recover a qualitative 3D representation of the world. We leave this direction for future works.
>
> ***Q: "How was the number K=10 chosen? What happens if K=100, or K=1000? More generally, what is limiting the detail level of the fits?"***\
> A: In a flavor similar to [66] and [53] which use $K\leq20$, our main objective is to discover the most predominant geometric parts of a scene. Looking at DTU scenes, our intuition is that the minimal number of parts that coarsely explain the entire scene geometry is actually relatively small. Therefore, we arbitrarily cap the number of primitives to K=10. The current limiting factor for increasing the detail level of the fits is the memory consumption. In particular, having K=1000 implies optimizing 1000 texture images of relatively high resolution which does not hold in memory. In the following table, we report DTU results for different values of K. We can observe that increasing K improves the performances; we will add a discussion in the supplementary material.
>
> | Model | #P$\downarrow$ | CD$\downarrow$ | PSNR$\uparrow$ | SSIM$\uparrow$ | LPIPS$\downarrow$ |
> |-|:-:|:-:|:-:|:-:|:-:|
> | $K = 10$ (default) | 4.60 | 3.63 | 20.5 | 73.5 | 23.9 |
> | $K = 25$ | 7.00 | 3.58 | 21.0 | 74.6 | 22.5 |
> | $K = 50$ | 9.26 | 3.52 | 20.9 | 74.7 | 22.8 |
>
> ***Q: "Why does the model get caught in bad local minima (in more detail)? Would just increasing the number of restarts avoid the minima? [...] Why doesn't the soft rasterization and the transparency handling smooth out the loss landscape sufficiently?"***\
> A: More precisely, a typical bad local minima is one where some parts of the geometry are not reconstructed. This is mainly caused by the absence of primitives in this region at the random initialization. Increasing the number of restarts will solve this issue, however this is not ideal. To avoid these restarts, we tried adapting the smoothness parameter sigma of the soft rasterization or equivalently applying a Gaussian kernel to blur the image, but the experiment was not conclusive. Blurring the image allows pixels to be influenced by larger regions in the image, however it introduces approximations and ambiguities that cannot be easily resolved. We will add a discussion in the paper.
>
> ***Q: "Are the parameters (sigma) of the soft rasterization adjusted on a schedule during optimization? What is the effect, if any, of the soft rasterizer approximation on the optimization (i.e., what is the effect of varying sigma)?"***\
> A: As stated in our supplementary material L61 and L66, the smoothness hyperparameter $\sigma$ is set to the default value $10^{-4}$ during most of the optimization (20k iteration), then it is decreased to $5 \times 10^{-6}$ during the last 5k iteration. We found this second training stage to help compute pixel-accurate shape boundaries and improve the quality of the texture images. Using higher values of $\sigma$ makes the results blurrier.
>
> ***Q: "How does the model behave for different values of the parsimony parameters? Is there an ablation showing the effect of that parameter?"***\
> A: In our ablation study in Table 2, we evaluate the impact of removing the parsimony loss. In the following table, we report additional DTU results for other values of $\lambda_{\text{parsi}}$. We can observe that it directly influences the effective number of primitives found. When $\lambda_{\text{parsi}} = 0.1$, the regularization is too strong (roughly 1 primitive found) and it dramatically decreases the reconstruction performances. When $\lambda_{\text{parsi}}$ is much smaller, the number of primitives increases without significant improvements.
>
> | Model | #P$\downarrow$ | CD$\downarrow$ |
> |-|:-:|:-:|
> | $\lambda_{\text{parsi}} = 0$ | 8.86 | 3.65 |
> | $\lambda_{\text{parsi}} = 0.001$ | 7.44 | 3.61 |
> | $\lambda_{\text{parsi}} = 0.01$ (default) | 4.60 | 3.63 |
> | $\lambda_{\text{parsi}} = 0.1$ | 1.30 | 6.88 |

---

> > ### Comment · Reviewer_3Dgw · 2023-08-11
> > **Thanks for the rebuttal**
> >
> > Thanks for the rebuttal and the new experiments. I have to say I'm still concerned over the results of increasing the number of primitives. If I understand the results correctly, #P can roughly double without CD or PSNR changing much (#P 4.6 -> 8.86, CD 3.63 -> 3.65, or in upper result, #P -> 9.26, CD -> 3.52). This suggests to me that there is still some issue with the optimization or the loss.
> >
> > It could be interesting to ask a human artist to approximate some of the simpler DTU 3D shapes with blocks as a ceiling on the performance of this method. A human-authored result could also be used as an initialization or a test to make sure the optimization does not wander away from a good solution (given the results here and what I understand about the soft rasterizer, it is not clear to me that the loss is reliably minimized at a known good solution).

---

> > > ### Author Response · Authors · 2023-08-12
> > >
> > > Thank you for the follow-up comment. For DTU scenes which have few simple geometric parts, modeling more primitives does not significantly increase performances because our default approach is already capturing most of the scene geometry. We observe in the qualitative results that modelling more primitives in this case typically leads the model to reconstruct one block with two (or more) side-by-side blocks, which does not improve reconstruction. Improving significantly the reconstruction for these scenes would imply modeling the fine geometric details, like the stairs-shape of the house roof or the non-superquadric pumpkin shape. Note that for more complex scenes like BlendedMVS where parts can be missed, like the gundam scene, allowing more primitives is what helps finding a good solution (Fig 4, bottom 2 rows).
> > >
> > > A human baseline comparison would indeed be interesting. However, it is not trivial to do in practice, since 3D modeling requires a strong expertise with specific tools, and there is no tool that we know of that would enable a user to select a superquadric shape in an intuitive way.

---

> > > > ### Comment · Reviewer_3Dgw · 2023-08-16
> > > >
> > > > I suppose it depends on what you mean by "most," but there seems to be obvious room for improvement in the fidelity of these results relative to the true shape. For example, the DTU house is reconstructed by only two big blocks, ignoring the slope of the roof entirely (not just the stairs-shape of the building facade). An off-angle view of that reconstruction would probably not look good.
> > > >
> > > > Again, I think some amount of under-fitting may be acceptable, but as a reader I would like to know more about *why* this under-fitting occurs. Adding more failure cases as suggested by other reviewers could help.

---

> > > > > ### Author Response · Authors · 2023-08-18
> > > > >
> > > > > Thanks for the follow-up. We will include a failure case figure with several examples, along with a discussion describing each case, as suggested by xDAr. For the specific case of the house, the renderings indicate that the roof under-fitting likely comes from the fact that the texture of the roof is uniform and, because there is no strong off-angle views in the training data, all training views are quite well reconstructed from a relatively coarse geometry.

---

### Official Review · Reviewer_xDAr · 2023-07-01

**Soundness:** 3 good
**Presentation:** 4 excellent
**Contribution:** 4 excellent
**Rating:** 7
**Confidence:** 4

**Summary:**

This paper presents an optimization-based method to recover 3D textured primitives from image observations. The core of the method is a fully differentiable and parsimonious 3D scene representation consisting of a background dome, ground plane, and a collection of multiple superquadrics, where the transformations as well as the UV-mapped textures are jointly optimized. By introducing transparency into the primitives, the authors devise an effective rendering mechanism that is optimizer-friendly and adaptive to different numbers of primitives. The proposed method is compared to 3D-based counterparts that fit primitives to the ground-truth point cloud and shows better representational compactness and fitting tightness. The primitive-based representation allows for various applications such as physics simulation and amodal shape completion.

**Strengths:**

1. The task of 3D decomposition from image collections is novel. Previous works either fit 3D primitives directly from point clouds, or build a low-level representation of the scene from images. This paper is the first effort to compute 3D primitives using image observations, and such a task would be useful in many other aspects including data transfer, scene editing, etc.
2. The idea of optimization through rendering is novel. Inspired by the effective differentiable rendering scheme from NeRF, the paper introduces the transparency of the primitives and uses alpha blending to render, which is fast and effective. Such an idea is clever and inspiring.
3. The presentation is good and the writing is clear. The paper is very clear in explaining the core ideas and the scene representation. Also, I could not find any grammatical errors or obvious typos in the writing.
4. Results are compelling. The decomposition of 3D primitives is promising even for very complicated shapes such as robots. The decision boundaries of the primitives are clear and could be even useful for unsupervised semantic segmentation.

**Weaknesses:**

The main weakness of the paper is in its evaluation. In particular, the following experimental results should be reported:
1. Running EMS [41] and MBF [55] on the geometries recovered by, e.g., NeuS [13, 67]. From Fig.3 it could be clearly seen that even for 'GT point cloud' provided by DTU dataset, there are missing regions. Hence it would be fairer if the input of EMS and MBF is point clouds sampled from the NeuS-mesh. This would make the experimental results & comparisons more convincing as this aligns the inputs of the proposed method and the baselines.
2. The rendering quality (e.g., PSNR) should be compared against NeRF-based baselines, or other mesh-based baselines, such as NVDiffRast, PartNeRF, or Nerfacto. Although I could understand that the rendering quality is not the main pursuit of the paper, it would be good to have such a comparison for completeness. Actually, the author could add metrics that compare the representation compactness (such as the file size for a scene) just to show off the advantage of using this mid-level representation. Additionally, it would be very interesting to show how the rendering quality, optimization time, and scene size vary when the number of primitives grows.
3. Running time of the method and the baselines should be compared (or at least discussed).

**Questions:**

1. If the superquadrics are optimized to a cuboid, the textures along the edge will be squeezed and thus wasted (as it will not be reflected in the final rendering). Is there a possibility that the UV mapping could be also optimized to best utilize the pixels in the texture?
2. In Eq(2), how is $\Delta_j(\mathbf{u})$ computed in a differentiable manner? It would be helpful to visualize $\mathbf{O}$ (in supplementary maybe) and describe [7] in more detail for better understanding.
3. How could the method be extended to large scenes? Will a coarse-to-fine optimization strategy help solve the local minima issue?

**Limitations:**

The limitations are clearly stated in Sec4.3, and the author should include some exemplar failure cases (figures) in the supplementary material.

---

> ### Author Rebuttal · Authors · 2023-08-09
>
> We thank the reviewer for their valuable feedback. In the following, we address the main concerns raised in this review. We will incorporate relevant suggestions in the revised version.
>
> ***W1: Additional baselines using geometry from NeuS***\
> A: In the following table, we report DTU results for MBF and EMS using the geometry extracted from NeuS [67], instead of the GT point clouds. More specifically, we use the official repository and the official config files to optimize the model and extract a mesh, then use uniformly sampled 3D points as input for the baselines. Our approach still performs better than these baselines. We hypothesize that these baselines are worse than the ones using GT 3D point clouds because NeuS-mesh is extremely accurate in the object region but hallucinates geometry for unseen or ambiguous regions like the background; fitting primitives in these regions penalizes a lot the final scores. Note that these regions are in general filtered using GT object masks before reporting quantitative and qualitative results on DTU (e.g., see [74, 50, 73, 67]). We will add the results for all scenes in the paper.
>
> | Model   	| Input |  S24 |  S40 |  S63 | S105 | Mean CD | Mean #P |
> |-------------|:-----:|:----:|:----:|:----:|:----:|:-------:|:-------:|
> | EMS w/ NeuS |  Img  | 8.73 | 7.22 | 7.45 | 9.27 |   8.17  |   8.7   |
> | MBF w/ NeuS |  Img  | 4.22 | 3.77 | 3.14 | 6.31 |   4.36  |   53.7  |
> | Ours    	|  Img  | 3.25 | 1.16 | 3.40 | 5.21 |   3.25  |   4.5   |
>
> ***W2: Rendering comparison with NeRF***\
> A: We agree that having rendering metrics for a reference method would be good for completeness; we will add the results for NeRF in the supplementary material. In the following table, we compare PSNR for our approach and NeRF using the results reported in [73] on 4 DTU scenes. As expected, our results are clearly worse when evaluating rendering on viewpoints close to the ones used for optimization. However, our results (Fig. 5 first row) are much better for very different viewpoints.
>
> | Model |  S24 |  S40 |  S63 | S105 | Mean |
> |-|:-:|:-:|:-:|:-:|:-:|
> | NeRF  | 26.2 | 26.8 | 32.0 | 32.1 | 29.3 |
> | Ours  | 19.1 | 21.8 | 23.4 | 20.8 | 21.3 |
>
> ***W2: Results when the number of primitives grows***\
> A: In the following table, we report DTU results (effective number of primitives #P, Chamfer Distance and image rendering metrics) averaged over the 10 scenes and 5 random seeds for different values of K (the maximum number of primitives). We can observe that, as expected, increasing K improves the performances. Regarding the computational cost, increasing K does not change the overall runtime (roughly 4 hours for all experiments) but it impacts the memory consumption. We will add a discussion as well as visualizations in the supplementary material.
>
> | Model          	| #P$\downarrow$ | CD$\downarrow$ | PSNR$\uparrow$ | SSIM$\uparrow$ | LPIPS$\downarrow$ |
> |-|:-:|:-:|:-:|:-:|:-:|
> | $K = 10$ (default) | 4.60 | 3.63 | 20.5 | 73.5 | 23.9|
> | $K = 25$ | 7.00 | 3.58 | 21.0 | 74.6 | 22.5 |
> | $K = 50$ | 9.26 | 3.52 | 20.9 | 74.7 | 22.8 |
>
> ***W3: "Running time of the method and the baselines should be compared (or at least discussed)"***\
> A: We report our approach timings in Section 3.3. 'Optimization details' paragraph. As stated in L230, optimizing our model roughly takes 4 hours on a single GPU. We will add a discussion about the computational cost of the baselines. Since they operate directly on the 3D point clouds without computing textures, they are typically much faster and compute primitives in a couple of seconds. To get comparable timings, one would also have to account for the 3D point clouds computation from calibrated images, which is typically longer; for example, running COLMAP MVS on a DTU scene roughly takes 10 minutes.
>
> ***Q1: "Is there a possibility that the UV mapping could be also optimized to best utilize the pixels in the texture?"***\
> A: Indeed, the UV mapping could theoretically be optimized to best utilize pixels depending on the shape of the superquadric. However, this would imply additional regularization terms or another finetuning stage to prevent the mapping from diverging when the textures are randomly initialized. In addition, as stated L297 in our limitations section, our texturing model could also be improved to account for the primitive geometry. We leave these improvements for future works.
>
> ***Q2: "In Eq(2), how is $\Delta_j(\mathbf{u})$ computed in a differentiable manner? It would be helpful to visualize $\mathbf{O}$ (in supplementary maybe) and describe [7] in more detail for better understanding."***\
> A:  For a given pixel $\mathbf{u}$ and a face $j$, $\Delta_j(\mathbf{u})$ corresponds to the signed Euclidean distance. More specifically, as described in [40] and [7], the edges of the face are projected onto the image plane, and the Euclidean distance between pixel $\mathbf{u}$ and face $j$ corresponds to the Euclidean distance to the closest edge, which can be computed (almost everywhere) in a differentiable way. We will add clarifications in our supplementary material.
>
> ***Q3: "How could the method be extended to large scenes? Will a coarse-to-fine optimization strategy help solve the local minima issue?"***\
> A: Similar to NeRF, extending this method to large-scale scenes is not trivial. In particular, it would involve splitting the scene into multiple regions (e.g., in a spirit similar to Block-NeRF) and dealing with objects of different scales. We believe that a coarse-to-fine optimization could help with the latter, by gradually increasing the number of primitives and decreasing their initial scales.
>
> ***Limitations: Exemplar for failure cases***\
> A: We will add failure case examples in the supplementary material.

---

> > ### Comment · Reviewer_xDAr · 2023-08-15
> >
> > Thank you for your detailed rebuttal. I've read all the reviews and the author's response.
> > My attitude towards this paper is kept positive and I recommend 'Accept'.

---

### Official Review · Reviewer_85Rz · 2023-07-02

**Soundness:** 3 good
**Presentation:** 3 good
**Contribution:** 2 fair
**Rating:** 5
**Confidence:** 4

**Summary:**

This paper (DBW) presents a way to fit superquadric blocks to a scene, starting from multiple images with calibrated cameras. Unlike recent other methods, they don't start with 3D data in the form of point clouds or meshes. The compositional nature of the recovered scene is shown by manipulating the recovered shape on standard tools, simulating other physics-based effects, etc. Another feature is the recovery of textures along with the geometry which adds appearance to the composed scene. Results are shown on selected DTU scenes as well as on a few real-world images.

**Strengths:**

Compositionally recovering a scene as textured super quadrics is what the paper primarily achieves. This is being done directly using differentiable rendering without going through a 3D representation. The optimisation is achieved by different losses to ensure good fit to the observed shape. Recovering texture along with shape is also a plus.

**Weaknesses:**

I have some concerns about the presentation of the prior work leading upto this one. My understanding is thus:

- Tulsiani et al have 2 papers in CVPR 2017. One is on Differentiable Ray Consistency (DRC) and the other on Learning Shape Abstractions (LSA) ([66] in the present manuscript)
- The DRC's is the approach in my view that should motivate DBW presented in this paper. It presents an unsupervised (or self-supervised) way to fit geometry from images with camera matrices. The LSA paper fits cuboids to point clouds and is similar in that aspect of the work.
- Later, Superquadrics were reintroduced in CVPR2019 [53] and fitted probabilistically in CVPR2022 [41], both from point cloud data.
- I see the DBW paper as introducing Superquadrics to the DRC paper, taking ideas also from papers [41], [53], [66], and others.
- I am surprised that the DRC paper is neither acknowledged nor referred to. Is there a reason for that?

**Questions:**

Ablation results in Table 2 and in the supplementary are interesting. However, I see that all rows (except possibly the last row that removes L_perc) have results that are very close to one another. I realise only one loss term is removed in each of the table rows and I don't see a dominant factor in any. What happens if multiple losses are removed? I realise the combinatorics is high, but I feel we could have more insights from them

See my question on the DRC paper elsewhere.

It is true that given posed camera views (as necessary for this work), a 3D representation of the scene (as volumetric grids, point clouds, or meshes) is available effectively for free (using well-established methods). Why not use them also in the DWC framework? What happens if some 3D information is added?

**Limitations:**

The primary complaint I have about this work concerns its utility to tasks.

- Compositionality and interpretability are the major advantages of compositional segmentation of 3D scenes into parts. This holds when the geometry was fit into cuboids in the prior work.
- Replacing cuboids with superquadrics does not increase its compositionality or interpretability, in my view. The recovery shape may appear to fit the scene better, but which downstream tasks are enabled by it?
- It is good to see the scene being edited, collided/collapsed, water poured over it, etc. However, these are enabled at this level even by a cuboid decomposition.
- I am glad to see texture being recovered simultaneously; However, the quality of texture combined with the "boxiness" of superquadrics make the new views generated incorrect. This can be seen in the supplementary videos. For example, th Scan24 of the "house" appears boxy even with texture applied to it.
- NeRF and its descendants will provide much higher-quality view generation with appearance than this method can ever give as the geometry is approximated here. So, new view synthesis with the recovered texture is not really useful.
- Simple geometric reasoning of the scene is possible with compositional segmentation done by DBW, but I don't see a whole lot added by this work over doing the same with simple blocks or a simple extension of the DRC work.

I would like to know which use-cases are much better served by the DBW method presented here.

---

> ### Author Rebuttal · Authors · 2023-08-09
>
> We thank the reviewer for their valuable feedback. In the following, we address the main concerns raised in this review. We will incorporate relevant suggestions in the revised version.
>
> ***Weaknesses: "The DRC's is the approach in my view that should motivate DBW presented in this paper. [...] I see the DBW paper as introducing Superquadrics to the DRC paper [...] I am surprised that the DRC paper is neither acknowledged nor referred to. Is there a reason for that?"***\
> A: We agree that DRC [Tulsiani et al., 2017] is a pioneer analysis-by-synthesis method leveraging multiple views of a scene, thus related to our work, and we will add a discussion in the paper. However, it differs from our approach in three important ways. First, the problem setting is different: DRC is a learning approach for single-view reconstruction while ours can be seen as an optimization method for multi-view stereo. This is extremely important since DRC needs access to a training set made of multiple observations of scenes similar to the ones expected at inference, while our method is optimized on a single scene, without a training stage. Second, the 3D representation is different: DRC leverages a dense voxel grid while our approach optimizes textured parametric meshes. Third, the output is different: DRC predicts the 3D of the scene as a whole (a soup of voxels), whereas our method outputs multiple 3D elements that relate to different parts of the scene. This is typically not trivial and motivated fundamental works like [66] or [53], as well as applications (see Fig. 5).
>
> ***Q: Ablation results. "I see that all rows [...] have results that are very close to one another. I realise only one loss term is removed in each of the table rows and I don't see a dominant factor in any. What happens if multiple losses are removed?"***\
> A: Our ablation study removes one component at a time, which indeed leads to smaller differences with the complete method than if we were to combine the different rows. Note however that we ran each experiment 5 times, and reported the standard deviation for each experiment in the supplementary material, making it possible to identify which difference is significant. For example, the parsimony loss is essential to obtain a low number of primitives. The improvement in terms of Chamfer distance provided by TV, noise and curriculum are also significant. The overlap penalization indeed has a smaller quantitative impact, but improves the results qualitatively. As pointed out by the reviewer the combinatorics is high (5 runs for 6 factors ablated sequentially in any order would mean 5 * 6! =3600 experiments), but if the reviewer points to some combination that they feel would be relevant we will be happy to add them.
>
> ***Q: "Given posed camera views, a 3D representation of the scene is available effectively for free. Why not use them also in the DBW framework? What happens if some 3D information is added?"***\
> A: Our goal was precisely to develop a method which, opposite to prior works, did not rely on a 3D reconstruction on the scene. Indeed, as outlined in the paper, fitting primitives to a 3D representation comes with the burden of dealing with noisy and incomplete data. In particular, the 3D output for unseen regions would be empty or hallucinated (if not random as illustrated by our amodal synthesis experiment Fig. 5) and computing 3D-based losses in these regions does not make much sense. Our proposed approach to simultaneously solve for MVS and 3D decomposition helps dealing with this ambiguity. While we agree that adding to our framework the possibility to use 3D information in addition to images could be appealing (for example, increasing convergence speed), we believe it would not be trivial to do so in a way that is not penalized by noisy and incomplete 3D data.
>
> ***Q: "Replacing cuboids with superquadrics does not increase its compositionality or interpretability, in my view. The recovery shape may appear to fit the scene better, but which downstream tasks are enabled by it?", "I don't see a whole lot added by this work over doing the same with simple blocks"***\
> A: The novelty of our method is not to replace cuboids with superquadrics: the key novelty of our approach is to compute a 3D decomposition of a scene using image observations, instead of computing 3D primitives from point clouds like previous works. Our work could easily be applied with other types of primitives, and we actually started by reconstructing simple scenes such as the one of Fig. 1 with cuboids. On such scenes, the quality of the reconstruction was slightly worse with cuboids, but the same applications were indeed enabled. However, many other scenes (e.g., scene with fruits or the last 5 rows of Fig. 3) cannot be handled at all with simple cuboids. Considering superquadrics instead enabled the same applications for these scenes and not being restricted to "boxy" scenes.
>
> ***Q: "the "boxiness" of superquadrics make the new views generated incorrect", "NeRF and its descendants will provide much higher-quality view generation with appearance than this method can ever give as the geometry is approximated here"***\
> Indeed, our approach cannot be expected to compete with NeRF in terms of geometry or view generation accuracy for parts of the scene seen from multiple viewpoints and well reconstructed by NeRF (see answer to xDAr W2 for a comparison with NeRF). Our quantitative results for rendering aimed at analyzing the contribution of the different parts of our methods, not claiming state-of-the-art results. However, our view generation can be expected to be and actually is much better than the one provided by NeRF when looking at part of the scene seen in a single image or completely unseen, as showcased by our amodal view synthesis results (Fig. 5 first row)

---

### Official Review · Reviewer_ZNqf · 2023-07-06

**Soundness:** 3 good
**Presentation:** 2 fair
**Contribution:** 3 good
**Rating:** 5
**Confidence:** 4

**Summary:**

This paper focuses on parsing a scene into mid-level 3D representations made of a small set of textured primitives. The main contributions are as follows:1) Unlike existing primitive decomposition methods which requires 3D data, this approach relies directly on images as input.
2) The proposed method reconstructs and decomposes 3D models through differentiable rendering based optimization, by modeling primitives as superquadric meshes and optimizing their parameters from scratch with an image rendering loss. 3) A transparency is modeled for each primitive, which is critical for better optimization and also enables handling varying numbers of primitives.

**Strengths:**

This paper propose a novel 3D reconstruction, segmentation and texture mapping approach by a differentiable rendering based optimization, which is heuristic for 3D content generation. The technical part sounds correct and reasonable, and the experimental results show better quality to other SOTA methods of primitive fitting.

**Weaknesses:**

1. The motivation is this paper should be made clear. Why using a differentiable rendering based optimization? What is its advantages to other SOTA MVS reconstruction and texture mapping works.
2. Although lacking comparisons to SOTA MVS reconstruction and texturing works, the reconstruction quality of this approach seems not so good as these SOTA methods.
3. The proposed method is more similar to some evolution-based 3D reconstruction methods, like Hiep, V.H.; Keriven, R.; Labatut, P.; Pons, J.P. Towards high-resolution large-scale multi-view stereo. In CVPR 2009, pp. 1430–1437. Usually these kinds of methods consume much time for iterative evolution process. The paper lacks time statistics.
4. As claimed in the paper, the proposed method works better for water-tight objects and is not so suitable for scenes, which will limit its applications.

**Questions:**

1. Can more explanations be added in introduction for the motivation why we should use a differentiable rendering based optimization?
2. Can comparisons with some SOTA MVS reconstruction and texturing works be added?
3. Since this method is similar to a vectorized modeling work, can comparisons with some vectorization works like "PolyFit" be added?
4. Since this method is more similar to an evolution-based 3D reconstruction method, can comparisons with some evolution-based 3D reconstruction works like "Hiep, V.H.; Keriven, R.; Labatut, P.; Pons, J.P. Towards high-resolution large-scale multi-view stereo. In CVPR 2009, pp. 1430–1437" be added?
5. The time statistics of the experimental cases should be reported.

**Limitations:**

As claimed in the paper, the proposed method works better for water-tight objects and is not so suitable for scenes, which will limit its applications. Can you explain more clearly the reason?

---

> ### Author Rebuttal · Authors · 2023-08-09
>
> We thank the reviewer for their valuable feedback. In the following, we address the main concerns raised in this review. We will incorporate relevant suggestions in the revised version.
>
> ***W1, Q1: "The motivation in this paper should be made clear. Why using a differentiable rendering based optimization? What is its advantages to other SOTA MVS reconstruction and texture mapping works?"***\
> A: Optimizing geometry and texture through differentiable rendering is a single stage process that circumvents multi-stage procedures from traditional MVS methods (e.g., COLMAP), in a fashion similar to NeRF. Such an optimization stage is simple and intuitive and we believe it is a key advantage for further developments. Compared to SOTA MVS works, the main advantage of our approach is that it decomposes a scene into different primitives that are independently actionable. In particular, this enables new applications such as part-based physics simulation (e.g., Fig. 1c and videos in section B of the supplementary index.html), simple scene editing (e.g., Fig. 5 second row) and amodal completion (e.g., Fig 5 first row and section C of the supplementary index.html). We will add clarifications in the introduction.
>
> ***W2, Q2: "Can comparisons with some SOTA MVS reconstruction and texturing works be added?", "the reconstruction quality of this approach seems not so good as these SOTA methods"\
> W3, Q4:  "Can comparisons with some evolution-based 3D reconstructions works like Hiep et al. be added?"***\
> A: These works are indeed related to our work in the sense they tackle the problem of multi-view stereo. However, these works do not fulfill our primary objective to decompose a scene into distinct geometric parts. Nevertheless, we agree that having metrics for some reference MVS methods would be good for completeness; we will add reconstruction results for COLMAP MVS and a NeRF-based method in the supplementary material.
>
> ***Q3:  "Can comparisons with some vectorizations works like "PolyFit" be added?"***\
> Two potentially relevant papers are actually named PolyFit:
>
> 1) [Nan and Wonka, Polyfit: Polygonal Surface Reconstruction from Point Clouds, ICCV 2017] relies on fitting planes to a point cloud with RANSAC, and is thus similar in spirit to the MBF [Ramamonjisoa et al., ECCV 2022] baseline we report in the paper and one can expect the same kind of limitations. We tried running this comparison during the rebuttal period but did not manage to make it work. Specifically, the optimization froze for a couple of hours without responding, we will investigate further the reason for this issue.
>
> 2) [Dominici et al., Polyfit: Perception-aligned Vectorization of Raster Clip-art via Intermediate Polygonal Fitting, SIGGRAPH 2020] is an image vectorization paper, and we believe extending it to 3D data is not obvious.
>
> ***W3: "The paper lacks time statistics."\
> Q5: "The time statistics of the experimental cases should be reported."***\
> A: We report our approach timings in Section 3.3. 'Optimization details' paragraph. As stated in L230, optimizing our model roughly takes 4 hours on a single NVIDIA 2080 Ti GPU. We will add a small discussion about the computational cost of the baselines. In particular, since they operate directly on the 3D point clouds without computing textures, they are much faster and compute primitives in a couple of seconds. To get comparable timings, one would also have to account for the 3D point clouds computation from calibrated images, which is typically longer; for example, running COLMAP MVS on a DTU scene roughly takes 10 minutes.
>
> ***W4, Limitations: "As claimed in the paper, the proposed method works better for water-tight objects and is not so suitable for scenes, which will limit its applications. Can you explain more clearly the reason?"***\
> A: We are not sure where this is claimed in this paper, our approach works for multiple objects (many examples) and we report a few examples of BlendedMVS and NerfStudio scenes (Fig. 4 and supplementary material). Overall, because our approach involves meshes with clearly defined surfaces, it will indeed struggle to model transparent objects like glass or concepts like fog or clouds. We will make this clear.

---

> > ### Comment · Reviewer_ZNqf · 2023-08-21
> >
> > Thank you for your detailed rebuttal. I've read all your responses. My attitude towards this paper is to weakly recommend 'Accept', if you can provide comparisons with some other vectorization works like Nan and Wonka, Polyfit, and solved all the problems and improvements you promised in the rebuttal.

---

### Decision · Program_Chairs · 2023-09-21

**Decision:**

Accept (poster)

**Comment:**

The paper had mixed ratings: 1 accept, 2 borderline accept, and 1 borderline reject. Author rebuttals were effective in addressing the concerns of the reviewers. The negative reviewer (borderline reject) did not answer the rebuttal and participated in the discussion. The area chair agreed with the ratings of the positive reviewers and recommend acceptance.